# Postharvest Biology and Technology of Loquat (*Eriobotrya japonica* Lindl.)

**DOI:** 10.3390/foods12061329

**Published:** 2023-03-20

**Authors:** Hafiz Muhammad Shoaib Shah, Ahmad Sattar Khan, Zora Singh, Saqib Ayyub

**Affiliations:** 1Horticulture, School of Science, Edith Cowan University, 270 Joondalup Drive, Joondalup 6027, Australia; 2Postharvest Research and Training Centre, Institute of Horticultural Sciences, University of Agriculture, Faisalabad 38040, Pakistan

**Keywords:** browning, chilling injury, *Eriobotrya japonica*, fruit ripening, postharvest physiology

## Abstract

Loquat (*Eriobotrya japonica* Lindl.) fruit is a rich source of carotenoids, flavonoids, phenolics, sugars, and organic acids. Although it is classified as a non-climacteric fruit, susceptibility to mechanical and physical bruising causes its rapid deterioration by moisture loss and postharvest decay caused by pathogens. Anthracnose, canker, and purple spot are the most prevalent postharvest diseases of loquat fruit. Cold storage has been used for quality management of loquat fruit, but the susceptibility of some cultivars to chilling injury (CI) consequently leads to browning and other disorders. Various techniques, including cold storage, controlled atmosphere storage, hypobaric storage, modified atmosphere packaging, low-temperature conditioning, heat treatment, edible coatings, and postharvest chemical application, have been tested to extend shelf life, mitigate chilling injury, and quality preservation. This review comprehensively focuses on the recent advances in the postharvest physiology and technology of loquat fruit, such as harvest maturity, fruit ripening physiology, postharvest storage techniques, and physiological disorders and diseases.

## 1. Background 

Loquat fruit is believed to have originated during the Han Dynasty in southeast China approximately 2000 years ago [1]. Currently, it is cultivated in many countries around the globe, while China, the United States, Japan, Turkey, Spain, Pakistan, India, Brazil, Greece, and Cyprus are the top producers around the globe [2]. Loquat fruit is a rich source of vitamins, phenolics, minerals, sugars, organic acids, and antioxidants which protect the consumer against many diseases. Moreover, loquat fruit reduces the levels of free lipoproteins, which narrow the arteries and increase blood cholesterol. Flavonoids are only present in the peel tissues of loquat fruit, while in addition, phenolics and carotenoids are present in the fruit flesh [3]. The highest total phenolic contents (TPC) have been reported in ‘Mizauto’ fruit as compared to other cultivars [4]. Loquat fruit ripening is a complex process of different physiological and metabolic changes, mainly including: ethylene biosynthesis and respiratory modifications [5], colour modifications due to carotenoid biosynthesis [6], sugar and acid metabolism contributing to the variation in fruit sensory attributes [7] and changes in the lignin, polysaccharide, and pectin contents during fruit ripening resulting in cell wall modifications and fruit firmness changes during ripening [8]. These changes are associated with complex transcriptional elucidations and interlinked metabolic changes in loquat fruit. Carotenoid biosynthesis is the main constituent of fruit colour and fully mature loquat fruit may exhibit an orange to yellow colour due to the level and type of carotenoids [9]. Based on the flesh colour, loquat cultivars are divided into two main groups, including white and yellow- to orange-fleshed cultivars (Table 1).

Chilling injury (CI) is the main constraint in the extension of the cold (<5 °C) storage life of loquat fruit, which consequently shortens the marketing window [12,13,14]. The application of polyamine [15,16], methyl jasmonate (MeJA) [17], modified atmospheric packaging (MAP) [18], and low-temperature conditioning (LTC) [19] have been found effective to mitigate the incidence of CI. Purple spot is another physiological disorder that is characterized by purple colour spots with irregular and depressed areas on the peel of loquat fruit due to a calcium deficiency and a sudden change in the water potential of fruit at the colour break stage. However, purple spot symptomatic fruit has less acceptability in the supply chain. Gariglio, et al. [20] have reported that the incidence of the purple spot may be reduced by the application of calcium ethylenediaminetetraacetic acid, calcium chloride, and calcium nitrate before the colour break stage in loquat fruit. Moreover, direct sunlight exposure at the onset of colour break is also helpful to lower the purple spot incidence. Postharvest flesh browning incidence is an important constraint during postharvest handling of various fruits, including loquat, which is caused by cellular decompartmentalization. Loquat fruit is highly susceptible to the incidence of internal flesh browning during the postharvest supply chain. The application of 1-methylcyclopropene (1-MCP) [21], acetylsalicylic acid [12], oxalic acid [22], calcium chloride [23], MeJA [24], nitric oxide (NO) [25], and L-cysteine [26] has been reported to lessen the incidence of browning in loquat. Many other non-chemical alternatives, including controlled atmosphere (CA) storage [27], MAP [28,29], LTC [13], cold storage [30], and edible coatings [31] have also been reported to reduce the incidence of postharvest internal browning in loquat fruit.

Previously, the metabolic changes during fruit ripening and the postharvest quality management of loquat fruit have been summarized in different reviews [2,32]. Moreover, the mechanism and control strategies of major fungal diseases in loquat fruit have also been presented in a recent review [33]. However, comprehensive information explaining harvest maturity indices, harvest and handling at the farm, postharvest quality management techniques, postharvest disorders and their management, as well as major diseases in the loquat fruit were not discussed in detail. Hence, the present review focuses on detailed information on harvest maturity indices, fruit ripening, postharvest disorders and diseases, and the application of various postharvest technologies to maintain the quality of loquat fruit during the supply chain.

## 2. Determination of Harvest Maturity

In loquat fruit, many physiological and physical attributes have been used to determine the optimum harvest maturity in different regions of the world. However, fruit colour is the most adopted technique to determine the harvest maturity of loquat fruit, depending upon the cultivar [34,35]. Fruit aroma volatiles, total soluble solids (TSS), titratable acidity (TA), and the TSS/TA ratio have also been reported to estimate the optimum maturity of loquat fruit [36]. The level of sugars has also been reported to estimate the harvesting stage of loquat, as there is a higher concentration of fructose and glucose at maturity stage as compared to the immature fruit [37]. The optimum sugar level in the loquat fruit at maturity is 10° Brix [38]. The harvest stage has a significant direct association with the postharvest physiology and storage potential of loquat fruit. However, harvesting before the fully ripe stage is advised by different scientists for commercial purposes, and this stage can be identified by the eating ripe stage along with the yellowish orange colour of loquat fruit. On the other hand, the bruising quality of loquat is directly associated with the degree of ripening, as fully ripened loquat is more susceptible to mechanical damage than fruit harvested at earlier stages [39].

Due to a higher demand in the markets, loquat fruit is advised to be harvested before 100% colour development, as it is a widely accepted commercial practice. The other reason to harvest at the early ripe stage is that the incidence of mechanical damage is very high in the fruit harvested at the fully ripe stage. There may be mechanical bruising and postharvest losses in loquat fruit when harvested at a fully ripe stage due to soft fruit flesh [40]. However, there may be an unpleasant fruit taste if loquat is harvested at an immature stage. Consumer acceptance is the main indicator that should be considered during the harvesting of fruit. Harvest maturity has been found to significantly influence postharvest bruising incidence and consumer acceptability. ‘Algerie’ loquat fruit, harvested with yellow-orange colour stage with 10° Brix TSS, exhibited the best eating quality [34].

## 3. Fruit Physiology Modifications during Ripening

Loquat fruit ripening is the sum of some structural and biochemical changes, including an increase in fruit size and weight, modifications in fruit firmness, colour development due to carotenoid biosynthesis, and a decrease in the levels of organic acids [41]. Hamauzu, et al. [42] investigated the physiological and compositional changes in ‘Mogi’ loquat at different maturity stages. Ethylene production, respiration rate, flesh firmness, surface colour, TPC, organic acids and sugar content were observed in fruit harvested at different maturity stages. They observed that harvesting at the faded green colour stage resulted in a decline in fruit firmness, decreasing the sugar and organic acid contents. Malic acid was found to be the most abundant organic acid in loquat at maturity, although there was also a considerable amount of succinic, fumaric, and citric acids. There was a simultaneous increase in ethylene biosynthesis at maturity with a considerable decline in respiration rate and this was the optimum stage for harvest.

The relationship between loquat fruit ripening and ethylene biosynthesis is somewhat inconclusive, as there have been varied assessments made by different researchers. However, it is believed to be non-climacteric in nature due to the lack of a sharp rise in ethylene biosynthesis and respiration rate at harvest maturity [42,43,44]. Moreover, fruit maturation is independent on exogenous ethylene application [45], and irregular patterns of relationships between respiration and ethylene biosynthesis are not evident in loquat fruit despite being reported for many climacteric fruits [46]. Three genes, including *EjACO1*, *EjACO2,* and *EjACS1* from ‘Luoyanqing’ loquat fruit, have been cloned at various ripening stages and shown to account for endogenous ethylene production. The expression of *EjACO2* was observed in petals and leaves, whereas *EjACO1* and *EjACS1* appear in fruit. The ethylene production peaked at the colour break stage, where the *EjACO2* expression pattern was consistent with ethylene biosynthesis [47]. Respiration is another important factor associated with loquat fruit ripening and is catalyzed by ethylene biosynthesis. The rate of respiration during maturation is also correlated with the cultivars. The lowest respiration has been reported in ‘Centenaria’ loquat while ‘Mizauto’ and ‘Mizumo’ loquat exhibited a higher respiration rate up to two days after harvest, and declined thereafter [48]. However, the pattern of respiration and ethylene biosynthesis is still unknown in many cultivars and requires future investigations.

Loquat fruit undergoes colour modifications, which is an important maturity index based on the type of cultivar (yellow or white), contributed by the endogenous carotenoids production. The major constituent of ripe loquat fruit is *β*-carotene in the pulp, while *β*-cryptoxanthin has been found in the peel tissues [49]. There is a consistent increase in carotenoid biosynthesis with the advancement in the maturation of loquat fruit [50]. The advancement of fruit growth causes a reduction in organic acids, except for malic acid, which is the main acid at the ripe stage, and carotenoids increase in pulp tissues, which leads to the colour development in loquat fruit. During ripening, fruit softening increases due to the degradation and hydrolysis of pectin and lignin contents. Immature fruit exhibits a higher activity of phenylalanine ammonia lyase (PAL) enzyme, which catalyzes lignin biosynthesis in immature fruit. There was a gradual decline in PAL enzyme activity with the advancement in fruit maturation, leading to less lignin deposition in the cell walls of loquat fruit, resulting in fruit softening [51]. Cell wall depolymerizing enzymes play a significant role in the downregulation of fruit firmness with the advancement in fruit development. Pectin methyl esterase and polygalacturonase are the principal enzymes leading to the demethylation and misfunctioning of cross-linkages between the cell wall components, increasing the hydrolysis of pectin molecules in the cell wall. Similarly, the cellulase enzyme deteriorates the cell wall polysaccharides and causes fruit softening during fruit maturation [52].

Sugars and organic acids contribute to the fruit flavour and texture at the ripe stage. The level of sorbitol is at its maximum in unripe loquat fruit and decreases with the advancement of fruit maturation. However, sucrose accumulation is very high during the early fruit maturation phase and is also the predominant sugar in mature loquat fruit [2]. Similarly, glucose and fructose contents also increase with the advancement of loquat fruit maturation [7]. During ripening, the accumulation of sucrose by the metabolism of polysaccharide molecules is triggered by a series of sugar-metabolizing enzymes including sucrose phosphate synthase (SPS), sucrose synthase (SS), and acid invertase (AI), while the major enzymes catalyzing the metabolism of sorbitol in ripening loquat are sorbitol-6-phosphate dehydrogenase (S6PDH), sorbitol dehydrogenase (SDH), and sorbitol oxidase (SOX) [53,54,55]. The TSS/TA ratio contributes to the sensory attributes of loquat fruit, including flavour and texture, while approximately seventy-eight various aroma volatiles have also been identified in loquat fruit [56]. Although the primary aroma volatiles have also been reported in the literature, changes in the levels of conjugated and non-conjugated aroma volatiles have not been determined in different loquat fruit cultivars and warrant further investigation.

## 4. Harvest and Handling

Harvest and handling are directly associated with the postharvest quality of loquat, as about 50% of losses are due to the harvest and handling operations of the fruits. Mostly, the fruit is harvested at 100% colour development for the local markets. In Cyprus, the fruit is mainly harvested at an early stage (50–75% colour development) and exhibited better shelf life than late harvesting at 100% colour development [36]. The harvesting stage of loquat fruit may be judged based on certain physical, physiological, and biochemical attributes (Table 2). Loquat fruit is harvested manually, using gloves to avoid contamination of the fruit with germs. Fruit should be cut from the nearby stalk and should not be pulled to avoid skin bruising, and then gently placed with care in the container. Rough handling causes bruising and bursting of the peel with sap flow, where later fungal and bacterial growth leads to deterioration of the fruit [57]. The best time for harvesting loquat is the morning after sunrise because there is less ethylene production, respiration rate, and lower fruit internal temperature, which can enhance the shelf life and postharvest quality of the fruit [58]. After harvest, fruit should be packed in a container with shock absorber material to avoid bruising during transport to the packhouse [59]. Package house practices include many operations, such as the grading of fruit based on colour and quality free from decay, the removal of insect pest symptoms, exposing the fruit to a low temperature, throwing out decayed fruit, and packaging. Shading for harvesting fruit is necessary with proper aeration to remove field heat, followed by transport to the packhouse. Packhouse operations start with the removal of fruit from containers after the sorting of fruit based on decayed and healthy fruit. Fruits are mainly graded based on their size and colour, free from any physical damage, bruising, and disease symptoms. Pre-cooling should also be practiced for fruit preservation because it reduces postharvest losses by reducing the ethylene and respiration rate [41]. Physical damage is the first constraint in fruit deterioration that initiates a stress condition, leading to a subsequent increase in malondialdehyde (MDA) content, polyphenol oxidase (PPO) enzyme, ethylene, and respiration rate, with a browning index [60]. Stem cutting with a clipper or sharp knife can help reduce physical damage. Possible operations from harvest to consumer are listed in Figure 1.

## 5. Postharvest Handling and Storage Techniques

### 5.1. Heat Treatment

Heat treatment (HT), conducted either alone or in combination with other techniques such as cold storage, MAP, and chemical application, has been shown to be effective for the quality maintenance of loquat fruit [14]. Hot air treatment is an effective technique to control CI in loquat fruit (Table 3). Hot air treatment at 38 °C for 5 h, followed by cold storage at 1 °C for 35 days, is very effective against chilling stress in ‘Jiefangzhong’ loquat fruit. The above study also showed a significant decline in electrolyte leakage and MDA content with reduced activities of the lipoxygenase (LOX) and phospholipase D (PLD) enzymes. Further, linoleic and linolenic acid levels were maintained, which delayed the production of oleic, stearic, and palmitic acids, in turn leading to a higher unsaturated-to-saturated fatty acid ratio [62]. Similarly, loquats exposed to hot air treatment (45 °C) for 3 h maintained a higher saturated/unsaturated fatty acid ratio, and inhibited membrane leakage by reducing the levels of MDA and hydrogen peroxide (H_2_O_2_) radicals [14]. Further, HT delayed activities of polygalacturonase (PG), PPO, peroxidase (POD), and PAL enzymes, maintained higher juice content, and lowered fruit decay incidence in loquat fruit. Moreover, fruit firmness was maintained due to the higher pectin and lignin content in the peel of HT-treated loquat fruit [63]. Similar hot air (35 °C, 5 h) treatment in cultivar ‘Jiefangzhong’ was useful in maintaining higher activities of ascorbate peroxidase (APX), catalase (CAT), and superoxide dismutase (SOD) with condensed POD enzyme, H_2_O_2_ content, and superoxide anion leading to lesser LOX activity and MDA production, with higher juice content and alleviation of fruit firmness [64]. Combined hot air and hot water dip applications before cold storage in ‘Jiefangzhong’ efficiently delayed the senescence of the fruit with a delayed reduction in TSS and TA, reduced decay, browning index, rot incidence, and improved sensory qualities with higher juice content [65]. The efficiency of hot air (37 °C) for 3 and 6 h, followed by cold storage for 60 days, demonstrated that heat application for 3 h was very helpful in lowering the internal browning of ‘Fukuhara’ loquat fruit pulp compared to 6 h with lower activities of PAL, POD, and PPO enzymes, which were negatively correlated with HT, and an intermittent exposure for 3 h with no effect on fruit firmness [66]. Similarly, hot air exposure at 48–52 °C, with cold storage at 2–5 °C, reduced CI symptoms, plasma membrane permeability, and respiration rate, and maintained higher activities of CAT, POD, and PAL enzymes. Heat shock-induced defence mechanisms ultimately alleviated chilling stress and respiration rate while enhancing the activities of CAT and POD enzymes and reducing the activity of the PAL enzyme [67].

### 5.2. Postharvest Chemical Treatments

Several chemicals have been used as potential inhibitors against senescence and physiological disorders to extend the storage life of loquat fruit (Table 4). 1-MCP is an ethylene inhibitor that is used for quality management and extending the shelf life of many horticultural commodities [73,74,75,76,77]. 1-MCP has been found to be very effective against internal browning and fruit decay incidence, with better sensory attributes in ‘Luoyangqing’ loquat during storage as well as maintained membrane integrity, reduced oxidation of polyphenols, superoxide anion accumulation, and activities of PPO and LOX enzymes [21]. Similarly, 1-MCP application lowered fruit decay in ‘Qingzhong’ loquat by inhibiting reactive oxygen species (ROS) production due to effective 1,1-diphenyl-2-picrylhydrazyl (DPPH) radical scavenging activity and lower activity of the PPO enzyme [78]. The application of 1-MCP before cold storage alleviated CI symptoms with a considerable decrease in electrolyte leakage, H_2_O_2_, superoxide radicals, and MDA content by maintaining lower activities of phospholipase C (PLC) and LOX enzymes and higher CAT enzyme activities [79]. The application of 1-MCP treated ‘Fuyang’ loquat fruit maintained a higher unsaturated to saturated fatty acid ratio, leading to higher membrane integrity and reduced CI symptoms. Moreover, 1-MCP treated fruit reduced the levels of cellulose, hemicellulose, and cyclohexanediaminetetraacetic acid (CDTA) soluble pectin than control [80]. The relationship between cell wall metabolism and CI is the key factor in the chilling susceptibility of loquat fruit and is mainly influenced by upstream activities of the phenylpropanoid pathway, including the activities of coenzyme A ligase:4 coumarate (4CL), the endomembrane-bound cinnamate 4-hydroxylase (C4H), and PAL enzymes. Exogenous application of 1-MCP suppressed the activities of PAL, C4H, and 4CL coenzyme A and maintained higher PG enzyme activity, leading to less pectin and lignin deposition in cold-stored loquat fruit and hence lowered the CI index [81].

MeJA, an ester of jasmonic acid, is a cyclopentanone compound known to regulate plant growth and is effective against physical and physiological stresses [110]. Recently, the role of MeJA in the alleviation of CI and quality deterioration in horticultural commodities has been examined [17,98,102]. MeJA induces chilling tolerance, mainly by inducing gene expression in *EjbHLH14*, *EjHB1*, and *EjPRX12* genes, which lowers lignin biosynthesis [17]. The mechanism of MeJA against chilling stress is mainly due to the activation of oxidative enzymes such as CAT, SOD, POD, and APX, which are mainly initiated by the higher H_2_O_2_ level and delay in the oxidation of unsaturated to saturated fatty acids during storage. Exogenous MeJA application lowered the ethylene biosynthesis, respiration rate, and PAL and PPO enzyme activities with higher DPPH-radical scavenging and SOD enzyme activities [96]. Moreover, there were a delay in H_2_O_2_ and superoxide anion production that inhibited LOX enzyme activity and elevated activities of APX, CAT, and SOD enzymes, leading to higher membrane integrity due to an increase in unsaturated fatty acids, hence inducing chilling tolerance in MeJA-treated ‘Fuyang’ loquat than control [97]. MeJA treatment inhibited the activity of the PAL enzyme and increased the activity of the PG enzyme, which lowered ROS production and fruit firmness due to a reduction in lignin, pectin, hemicellulose alcohol residues, and cellulose elevation, thereby inducing CI tolerance in ‘Fuyang’ loquat during cold storage. Additionally, CDTA- and water-soluble pectin were reduced with improved solubility of polysaccharides in the cell wall [98].

NO is the major bioactive signaling compound during abiotic and biotic stresses in plants that triggers a defense response and itself is the major scavenger of ROS during CI [111]. The inhibition of endogenous NO accumulation is the main factor that makes a fruit susceptible to oxidative stress and higher membrane leakage due to the activities of oxidative enzymes such as PPO and POD, with higher lignin deposition due to PAL enzyme activity. Alternatively, a 50 mmol·L^−1^ sodium nitroprusside (SNP) dip treatment maintained a lower activity of LOX and MDA content, leading to a reduction in lipid peroxidation with higher activities of APX, CAT, and SOD enzymes in ‘Dawuxing’ loquat, thereby maintaining cell membrane integrity and the suppression of POD and PPO enzymes. Additionally, the confirmation of the NO role in low-temperature tolerance and the signal transduction of jasmonic acid (JA) was confirmed by the exogenous application of 2-(4-carboxyphenyl)-4,4,5,5-tetramethylimidazoline-1-oxyl-3-oxide (cPTIO), an NO scavenger, and the NO inhibitors NaN_3_ and *L-NAME* in ‘Zaozhong No. 6′ young loquat fruit [112]. Chilling stress increased with the application of a NO inhibitor and scavenger with suppressed activities of SOD, POD, and CAT enzymes while MDA and H_2_O_2_ contents were aggravated. The application of cPTIO not only stimulated CI symptoms but also lowered endogenous NO biosynthesis during cold storage with reduced membrane integrity [25].

Sulphur is an essential nutrient required for plant growth and development and plays a crucial role in various physiological processes. It has been shown that improved plant resilience is directly connected to the increasing levels of sulphur dioxide (SO_2_) uptake and SO_2_ metabolism in plants. The application of SO_2_ helps to maintain the equilibrium between the production of ROS and their detoxification, which in turn helps to suppress ROS formation and prevent the lignification of the flesh of loquat fruit [33]. The treatment of loquat fruit with an SO_2_-releasing agent (SO_2_-releasing pad) substantially reduced the prevalence of decay and browning. The TSS and TA levels were observed to be higher than in control fruits [113]. Similarly, after 35 days of storage, loquat fruit treated with SO_2_ exhibited acceptable quality and a significantly lower reduction in total acidity and percentage of juice [15].

Several studies have reported that short-term N_2_ treatment delayed fruit ripening by reducing respiration and ethylene production [114,115]. A substantial delay in the rise in fruit decay rate, less reduction in TSS and TA, improved eating quality, and extended shelf life were observed in loquat fruits treated with short-term N_2_ for 6 h at 20 °C. Furthermore, increases in membrane permeability, MDA concentration, and superoxide anion generation rate were all significantly slowed by short-term exposure to N_2_. SOD, CAT, and POD activities were all substantially increased in N_2_-treated fruits compared to control fruits, which led to a decrease in CI [109].

Ozone is an allotropic species of oxygen that has powerful disinfection capabilities; as a result, its use in the postharvest processing of fruits is becoming more widespread. Under the right conditions during the production process, ozone will activate mechanisms that are involved in the protection of cellular structures against oxidative stress. This will increase the concentration of vitamin C, and consequently, the product will have more antioxidant properties [116]. Loquat fruit, treated with ozone, showed significantly less rise in lignin content and cell membrane permeability, as well as a decreased fruit deterioration rate [117].

Calcium is a very important nutrient that not only stimulates the growth of plants but also enhances the defense mechanism against many physiological and physical stresses [118]. It has been widely used to alleviate CI in cold-stored loquat fruit by triggering the suppression of ROS accumulation through the Helliwell-Asada pathway. Higher activities of dehydroascorbate reductase (DHAR), monodehydroascorbate reductase (MDHAR), glutathione reductase (GR), APX, CAT, and SOD were also exhibited by the calcium chloride (CaCl_2_)-treated fruit [90]. Calcium is also effective in inhibiting flesh and peel browning [105]. CaCl_2_ pre-storage treatment induced *EjCAMTA5*-mediated transcriptional regulation of *EjPLC6*-like and *EjLOX5* genes, which delayed membrane lipid hydrolysis and maintained cell membrane integrity, preventing loquat fruit browning during chilling stress [119]. The exogenous application of calcium compounds reduced membrane permeability, suppressed ethylene production and the respiration rate in loquat fruit, resulting in better fruit quality compared to the control [120]. Recently, melatonin has been found to be effective in the downregulation of MDA content and oxidative stress due to higher levels of DPPH, ferric ion reducing antioxidant power (FRAP), and 2,2′-azinobis (3-ethylbenzothiazoline-6-sulfonic acid) (ABTS) assays. In addition, higher phenolic content in melatonin-mediated loquat fruit was noted with higher activities of PAL, 4CL, C4H, and cinnamyl alcohol dehydrogenase (CAD) enzymes [104].

### 5.3. Edible Coatings

Edible coatings play a key role in the postharvest management of loquat fruit [31]. Carbohydrate-based edible coatings have been widely used to extend the shelf life of loquat fruit. The coating of loquat fruit with chitosan, a polysaccharide, maintained higher catechin and quercetin polyphenols while reducing ROS accumulation during cold storage [121]. Moreover, chitosan coating suppressed the respiration rate, ethylene production, internal browning, fruit firmness, and weight loss [122]. The application of a 1% chitosan coating in ‘Golden Nugget’, ‘Nespolone Rosso di Trabia’, and ‘Algerie’ loquat fruit was beneficial to delay changes in skin colour, TA, TSS, and sugar content, with lower fruit weight loss and internal browning, which were correlated with higher ascorbic acid (AsA) content, carotenoid, flavonoids, polyphenols, and reduced ROS production [123]. The packaging of loquat fruits with 3% myrtle leaves and exposing them to myrtle leaf vapour for 2 min inhibited weight loss, decay incidence, and fruit browning [124]. The total phenolic contents of loquat fruit, coated with an edible coating made from cactus pear mucilage, were higher than uncoated fruits, which indicated that the coating showed a beneficial effect in scavenging ROS, protecting cell membrane peroxidation, and delaying the senescence of loquat fruit [125]. Biomaterials with 0.5% *Nigella sativa* oil and 0.5% propolis extract reduced weight loss, maintained fruit firmness, and reduced fruit browning and decay in loquat fruits during cold storage [126]. The CI symptoms were downregulated in ‘Baiyu’ loquat fruit by the combination of a nano-silica and chitosan coating through maintaining higher cell wall integrity, leading to a subsequent decline in fruit weight loss and internal browning throughout the cold storage period with higher TSS, TA, and sugars [127]. Similarly, alleviation in CI incidence consequently reduced the activities of the LOX enzyme with lower MDA production and maintained membrane integrity by reducing the unsaturated fatty acids in fruit tissues by the chitosan and nano-montmorillonite composite film in loquat fruit [123]. Natural plant tree extracts from apricot and cherry were also effective in reducing microbial decay, bacterial colonies, and mycelia growth in loquat and strawberry fruits during postharvest storage. Moreover, there were no residual chemicals in coated fruits that may be harmful to consumer health [128]. Although the effectiveness of edible coatings has been reported in the literature, the production of fermented products is the major issue in the prolonged storage of edible coated loquat fruit [129]. Moreover, consumer acceptance toward coated fruit is the main hindrance to commercial usage of edible coatings for loquat fruit.

### 5.4. Cold Storage

Cold storage is widely used as it not only reduces the incidence of fruit decay and internal browning but also significantly extends storage life and maintains loquat fruit quality [127]. However, CI symptoms are shown to be elevated with low-temperature storage, which is associated with higher lignin biosynthesis in loquat fruit [27]. Moreover, some studies have shown that cold storage intensified CI with higher lignin deposition and juiceless flesh in cold-stored loquat fruit [85,98]. Tian, et al. [130] have outlined the minimum temperature to avoid chilling stress to loquats, which varies from 0 to 10 °C, depending upon the cultivars. They investigated chilling tolerance in ‘Wuxing’, ‘Zhaozhong’, and ‘Jiefangzhong’ loquat cultivars, observing the optimum temperature for these cultivars to be 1 °C, 8–10 °C, and 6–8 °C, respectively, for a maximum of 4 weeks of storage. However, cell wall integrity was maintained at lower temperatures (4 °C), while there was a higher PG correlation with higher temperature storage in ‘Karantoki’ and ‘Morphitiki’ loquat fruit stored at 20 °C [52]. Among cultivars, ‘Karantoki’ exhibited less phenolic content, antioxidative capacity, and acidity during storage. Similarly, there was an increase in fruit firmness throughout storage at 20 °C, with the increase in lignin content caused by the persistent rise in the activities of POD, CAD, and PAL enzymes in ‘Luoyangqing’ loquat fruit [13]. There was a higher rate of lignin deposition and firmness in fruit during the early days of cold storage while there was very little difference in loss of firmness after 3 weeks of storage [29]. The activities of sucrose hydrolysing enzymes were higher in chilling tolerant cultivars with higher activities of hexokinase and fructokinase enzymes involved in sugar generation signalling and hexose phosphorylation. Both of these enzymes maintained lower ethylene synthesis and respiration rates with better fruit quality after 35 days of cold storage [131].

### 5.5. Modified Atmosphere Packaging (MAP) and Controlled Atmosphere (CA) Storage

MAP is an effective postharvest strategy to enhance the shelf life and maintain loquat fruit quality by maintaining a higher CO_2_ and a lower O_2_ atmosphere around the fruit [130]. Several studies have outlined the efficacy of MAP in shelf life extension and quality management of loquats during postharvest storage (Table 5). The MAP storage of ‘Hafif Çukurgöbek’ loquat was beneficial in maintaining higher flavonoid, phenolic content, total sugars, organic acid, TA, and TSS with better skin colour and reduced internal browning and fruit decay at 5 °C for 30 days. There was significant inhibition in ethylene production in MAP stored over the control fruit [132]. Similarly, ‘Champagne de Grasse’ loquat fruit treated with 1-MCP followed by MAP storage at 5 °C for 4 weeks reduced ethylene production and maintained fruit quality [133]. Postharvest MAP storage in Xtend^®^ bags maintained higher levels of organic acids and sugars in ‘Morphitiki’ loquat fruit [18]. MAP, using biodegradable polymer polylactic acid bags, was effective in reducing internal browning, fruit weight loss, and the quality attributes of ‘Golden Nugget’ and ‘Algerie’ loquat fruit during low-temperature storage; however, there was an abrupt rise in the deterioration of fruit during the shelf period [134]. The MAP of biodegradable polymer polyactic acid on the postharvest sensory quality and storability of ‘Golden Nugget’ and ‘Algerie’ loquat fruits was effective in lowering internal browning, fruit weight loss, and quality attributes of both cultivars during low-temperature storage; however, there was an abrupt rise in deterioration of fruits during the shelf period [135]. ‘Ottawianni’ loquat fruit stored at 5 °C in active MAP with gas combinations (3% CO_2_ + 12% O_2_ +85% Ar) showed better quality attributes, along with a lower decay incidence as compared to the control [132].

CA storage is a specific gaseous composition around the fruit, with a higher concentration of CO_2_ and a lower O_2_. Atmospheric ethylene, CO_2_, O_2_, storage temperature and time, ripening stage, and commodity are key factors that determine the ethylene biosynthesis and respiration rates in CA storage. The CA storage with 10% O_2_ + 1% CO_2_ and 70% O_2_ for 24 h first, followed by CA storage with 10% O_2_ + 1% CO_2_ at 1 °C was effective in delaying fruit decay and maintained a TSS: TA ratio with higher endo-polygalacturonase (Endo-PG) and exo-polygalacturonase (Exo-PG) levels that also mitigated oxidative stress and internal browning [27]. It was observed that with higher O_2_ combined with polythene packaging, the activity of PPO was inhibited, resulting in lower oxidative stress and a browning index with better TSS and TA content in ‘Jiefangzhong’ loquat fruit [140]. The cost of CA storage and MAP bags and their unavailability in developing countries is the major limitation in their commercial use. Extensive research is required to develop the cost-effective and easily available MAP bags to lower the fruit quality deterioration and extend the shelf life of loquat fruit during postharvest storage and the supply chain.

### 5.6. Hypobaric Storage

Another very important storage technique that has been used to extend the shelf life by reducing senescence and delay in the ripening of loquat fruit is hypobaric storage. This technique is described by quick heat loss, reduced pressure, and oxygen levels during storage. Quality management and shelf life of loquat stored at 2–4 °C and 40–50 kPa for 7 weeks exhibited lower fruit flesh browning and fruit decay and higher juice contents with higher AsA content and TA [141]. The storage period can be increased up to 50 days when loquat is stored at 10 kPa pressure with low fruit decay and normal fruit flavour [142]. However, there is less information about the efficacy of hypobaric storage on the quality maintenance and postharvest storage life of loquat, which warrants further investigation.

## 6. Postharvest Physiological Disorders

### 6.1. Chilling Injury (CI)

Loquat is a highly perishable fruit that exhibits a very limited shelf life of about 10 days at ambient temperature due to excessive moisture loss and microbial decay. The desiccation of the fruit is a critical factor that determines the shelf life of any fresh commodity. Cold storage is a common way to lower the constraints related to ambient storage. However, some cultivars, such as the red-fleshed ‘Luoyangqing’ loquat cultivar, are very perishable due to CI, that can be determined by internal browning, juiceless texture, hardening of pulp, and stuck peel [143]. Yang, Sun, Wang, Shan, Cai, Zhang, Zhang, Li, Ferguson and Chen [82] determined the efficacy of different cultivars for CI and reported that ‘Baisha’ was the most susceptible among white-fleshed cultivars. There was increased lignification accompanied by the browning of pulp tissues and decreased juice content. Moreover, the activities of lignification enzymes, including guaiacol peroxidase (G-POD), CAD, and PAL were also increased with higher superoxide-free radical development. The incidence of CI is directly correlated with the abnormal metabolism of the fruit tissues due to increased PG and pectin methyl esterase (PME) enzyme activities [16]. Alterations in the levels of putrescine, spermidine, and spermine in fruit pulp have also been correlated with chilling stress and severe damage to loquat fruit [15]. Another physiological cause of CI might be a reduction in unsaturated fatty acids in the cell membrane of the fruit pulp due to oxidative stress followed by lipid peroxidation [144].

There is a direct correlation between the ethylene biosynthesis signalling and the expression of CI symptoms in loquat fruit. This role was demonstrated both in chilling sensitive red-fleshed ‘Luoyangqing’ and chilling tolerant white-fleshed ‘Baisha’ loquat cultivars [145]. Transcription records of both cultivars showed enhanced expression of *EjCTR1*, *EjETRS1b*, and *EjETR1* genes in both cultivars, while *EjEIL1* was expressed in ‘Baisha’ only at the middle stage of development; over the course of fruit ripening, there was a characteristic decrease in the expression of *EjEIL1* and *EjERS1a* genes in both cultivars. The development of the CI symptoms may be ascribed to oxidative stress, leading to the formation of MDA [97]. The mechanism in the CI resistance of loquat fruit was primarily due to higher levels of anti-oxidative enzymes, including APX, glutathione peroxidase (GSH-POD), SOD, and CAT activities that ultimately lowered H_2_O_2_ and superoxide radicals, leading to a significant decline in lipid peroxidation [146].

Another important physiological aspect which correlated with CI is lignin deposition in the cell wall of fruit tissues which causes the hardening of pulp and peel tissues. There are different mechanisms in both white- and red-fleshed loquat cultivars for lignin deposition [147]. Two genes, *EjMYB1* and *EjMYB2*, isolated from chill-stressed fruit, play important regulatory roles in lignification and promote the transcription rate for lignin synthesizing genes. Similarly, the role of *EjNAC1*, *EjNAC2*, and *EjODO1* genes in lignified stems and roots indicated that there was a decline in expression during the fruit development process [148]. Lignin biosynthesis increased due to increased *EjCL5*, *Ej4CL1*, and *EjPAL1* gene expression caused by trans-activation of the promoter with the overexpression of *EjODO1* [149].

#### Control of CI

Cold storage is the common technique to store loquat fruit, and some cultivars are susceptible to CI during low-temperature storage. LTC is an alternative technique for fruit quality management that regulates biochemical and physiological responses by suppressing CI incidence [150]. An antagonistic examination of LTC and HT based on the RNA-seq transcription database with the expression of *EjHSF* exhibited an abundance in *EjHSF1* transcription with a higher HSP *EjHsp* level in heat-treated loquat fruit, while there was a higher transcription of *EjHSF3* in LTC. Both *EjHSF1* and *EjHSF3* activated promoters of lignin biosynthesis-related genes and *EjHsp*. Therefore, *EjHSF* induced two distinct mechanisms in fruit regarding lignin deposition and transcription of *EjHSF3* induced in LTC was positively correlated with lignin biosynthesis, although *EjHSF1* induced chilling resistance in heat-treated loquat fruit [151]. The expression of genes involved in lignin deposition leading to CI was suppressed by LTC and HT, exhibiting a better quality of cold-stored loquat fruit [152]. LTC induces higher H_2_O_2_ accumulation at an early stage of loquat fruit storage and induces the chilling tolerance by regulating higher activities of antioxidant enzymes, including GSH-POD, APX, CAT, and SOD [153]. A significant reduction in CI symptoms with a double shelf-life extension was observed in ‘Luoyangqing’ loquat fruit exposed to LTC as compared to the control. LTC increased the resistance to browning in 60 days of cold storage, followed by a 2-day shelf life at 20 °C. Improved sensory attributes and low chilling damage were induced by LTC in loquat fruit [13].

Membrane leakage and internal flesh browning are two prominent symptoms of CI in loquat fruit. HT induces chilling tolerance in cold-stored loquat fruit by reducing the conversion of unsaturated fatty acids to saturated fatty acids, inhibiting the activities of phospholipase D (PLD) and LOX enzymes [62]. The upregulation of ascorbate and glutathione contents through the higher metabolism of sugars is another possible way to control the CI in HT-treated loquat fruit. HT induced higher ascorbate and glutathione contents and reduced fruit firmness, lignin deposition, and lipid peroxidation during low-temperature storage, leading to higher membrane integrity and chilling tolerance [14]. A comparative analysis of loquat fruit showing CI symptoms indicates a higher activity of the CAD enzyme, which is regulated through the expression of *EjCAD5*. Xu, et al. [154] reported that the application of hot air treatment suppressed the activity of the CAD enzyme and the expression of *EjCAD5* via a higher transcriptional expression of the *EjHAT1* promoter, hence maintaining less CI symptoms than control loquat fruit during low-temperature storage.

The exogenous application of anti-ripening chemicals is also effective to suppress CI symptoms in loquat fruit. The application of MeJA induces chilling tolerance by regulating the arginine metabolic pathway and increases γ-aminobutyric acid and proline contents in loquat fruit [102]. Moreover, MeJA increases the activities of antioxidant enzymes in cold-stored loquat fruit, leading to the reduced conversion of liquid to gel phase. MeJA-treated loquat fruit exhibited a higher metabolism of glutathione and ascorbic acid, followed by higher activities of SOD, CAT, and APX enzymes, and fewer CI symptoms [101]. 1-MCP reduces CI symptoms in loquat fruit by maintaining a higher unsaturated to saturated fatty acid ratio and reducing the activities of phospholipases and oxidative enzymes [79,80]. NO is also reported to lower the incidence of CI in cold-stored loquat fruit by regulating sugar metabolism [25].

### 6.2. Enzymatic Browning

Enzymatic browning is probably one of the most pronounced disorders in fruits and vegetables [76,155]. It reduces consumer acceptability of vegetables and fruits. The most accepted mechanism of browning is the oxidation of phenolic compounds by the oxidative enzyme PPO, which involves O-diphenol and oxygen oxidoreductase activities. Phenolic compounds are naturally present in the vacuole of plant cells while the oxidative enzymes are part of cellular cytoplasm [156]. Degradation and oxidative damage to the vacuolar membrane initiate the reaction of PPO with phenolic compounds. Monophenolase or cresolase enzymes catalyse the production of O-diphenol by the hydroxylation of monophenol, leading to further oxidation of O-diphenols to O-quinones. Both reactions are catalyzed by the PPO enzyme in the presence of O_2_. PAL and POD enzymes play a supportive role in the oxidation of phenolic compounds [157]. Potentially, SO_2_ fumigation was used for the inhibition of internal and pericarp browning in horticultural commodities [158]; however, due to its residual effects and carcinogenicity, it was prohibited in the fruit industry [26]. To avoid hazards caused by SO_2_ fumigation, many alternative chemicals, such as 1-MCP [21], acetylsalicylic acid [12], oxalic acid [22], calcium chloride [23], MeJA [97], and NO [25], have also been reported to minimize the incidence of postharvest internal browning in loquat fruit. However, internal flesh browning is still a major constraint in the loquat fruit supply chain and warrants an investigation of the strategies to control browning.

## 7. Diseases

Diseases cause fruit decay, limit preharvest production and postharvest life, and deteriorate fruit quality. A variety of pathogens are involved in pre- and post-harvest diseases; the main pathogenic organisms are *C. gloeosporioides*, *C. acutatum*, *Pseudomonas syringae*, *Phytophthora palmivora*, *Diplodia natalensis*, *Diplocarpon mespili* and *Botrytis cinerea*.

### 7.1. Purple Spot

Purple spot, caused by *Diplocarpon mespili* fungus, is a major preharvest disease that not only deteriorates fruit production in the preharvest phase but also limits the supply chain of loquat fruit in local and distant offshore markets [159]. Purple spot has been reported in many regions of the world and reduces about 50% of the commercial and cosmetic value of loquat fruit, and adversely impacts consumer acceptability. Early maturing cultivars are more vulnerable to being affected, and symptoms appear at the colour break stage during fruit development, leading to a decline in commercial value [46]. According to Gariglio, et al. [160] the initiation of purple spot is directly associated with a modified water relationship between peel and pulp tissues, leading to the spontaneous purple colour development followed by higher sugar accumulation in loquat fruit and a simultaneous rise in growth rate. Other symptoms are juiceless hard cells, collapsed cytoplasm with a breakdown in cellular composition out of the plasmalemma, flaccid peel tissues, and deformed fruit shape [46,161]. Sunlight exposure with low temperature and higher differences between day and night temperatures increases the purple spot incidence in loquat fruit. However, there is no influence of purple spot on the transpiration from the surface of the fruit [162]. Purple spot on loquat exhibit an irregular fruit surface and a more depressed area with slight spots of purple colour at the start of the disease, resulting in a 20% loss in natural fruit colour; however, this affects the epidermal tissues, with no effects on deeper flesh tissues. A calcium deficiency accompanied by a fungus attack may also lead to the development of deeper spots because there is a characteristic decline in the calcium level in loquat fruit throughout fruit development [162]. Contrarily, Blumenfeld [57] reported that the main cause of purple spots is the sudden change in water potential at the colour break stage of the fruit. This was very clear evidence that the purple spot appearance was correlated with dehydration of peel tissues and sudden exposure to low temperatures at the fruit colour break stage. Moreover, there was no difference in the permeability of the cuticle, and there was no damage to the cuticle layer of the fruit surface, although peel tissues were damaged by purple spot [44].

Another very important phenomenon is the reduction in the levels of Cu, K, and Fe in the peel tissues of the affected fruit compared to healthy fruit [159]. Purple spot incidence is mainly dependent upon the type of cultivar, the temperature change during fruit development, and the osmotic balance of the fruit. It has been demonstrated that early-maturing cultivars are more susceptible to the development of purple spot (60%) than late-maturing cultivars (30%) [163]. A significant increase in purple spots was associated with lower temperatures during the colour break stage, although the plants grown in a greenhouse that bear a higher night temperature compared to the open area, were less susceptible to purple spots. Direct exposure to the sunlight lessened purple spots on fruit exposed to light compared to shady areas. The management of purple spot is very important for the supply chain of loquat fruit. In this context, the efficacy of fruit thinning on the purple spot was assessed. There was higher carbohydrate accumulation with a higher growth rate in thinned fruit. Thinning led to a higher sugar content that was directly correlated with the incidence of purple spot.

In conclusion, there should be less or no fruit thinning to avoid the purple spot in loquat fruit [159]. In contrast, a higher accumulation of fructose, sucrose, glucose, and TSS lowered purple spot in the ‘Karantoki’ loquat. On the other hand, ‘Obusa’ exhibited the highest purple spot susceptibility due to the lower accumulation of sugars and TSS in fruit. Moreover, it was also proposed that there was a direct role of fruit maturity at the harvest stage on the development of purple spots, as there was the highest incidence in ‘Obusa’ harvested at commercial maturity. ‘Morphitiki’ was the most resistant cultivar against purple spots, as compared to ‘Karantoki’ and ‘Obusa’ [164]. However, there is a need to investigate the control strategies for the management of purple spot incidence.

### 7.2. Anthracnose

The incidence of anthracnose is caused by the fungus *Colletotrichum* genus [81,165]. There are many known species of genus *Colletotrichum*, such as *C*. *acutatum*, *C*. *eriobotryae, C. simmondsii*, *C. godetiae*, *C. gloeosporioides*, *C. siamense*, and *C*. *nymphaeae*; however, the most important are *C. gloeosporioides* and *C. acutatum* in loquat fruit, which deteriorate the fruit quality during postharvest handling [68,165,166]. The development of dark and water-soaked lesions are typical symptoms of anthracnose rot and often occur with pink gelatinous masses at the center of lesions during wet, moist, and warm temperatures. The severity of the disease is directly associated with a higher storage temperature and moisture content [94]. Different species of *Colletotrichum* were isolated in Taiwan [166]. In this regard, *β*-tubulin (TUB2)*, actin* (ACT), chitin synthase (CHS-1), glyceraldehyde-3-phosphate dehydrogenase (GAPDH), and internal transcribed spacer (ITS) genes were sequenced and two different types of *C*. *eriobotryae* and *C*. *nymphaeae* were found. However, the main pathogen causing anthracnose in Taiwan is *C*. *eriobotryae*. Oxidative stress and the production of ROS are mainly associated with the severity of the disease, as exhibited by higher H_2_O_2_ assay in infected fruit with reduced activities of SOD and CAT enzymes [166]. Similarly, morphological and phylogenic identification of a new strain, *C. godetiae*, in loquat fruit was identified by an internal transcribed space region (ITS) and four housekeeping partial genes relevant to TUB2, CHS-1, ACT, and GAPDH sequences by revealing reconstruction strain clusters through an ITS4 and ITS5 assay. Typical symptoms of *Colletotrichum* were similar concerning the molecular and morphological characteristics of this new strain, but it showed a different ITS assay [165].

Different strategies have been reported in the literature to control *Colletotrichum* infestations in loquat fruit. The exogenous application of MeJA has been reported to inhibit the spore germination and lesion diameter of *C. acutatum*. MeJA-mediated anthracnose rot, either by the direct inhibition of spores or indirectly through the higher H_2_O_2_ accumulation, resulting in reduced activities of PAL and PPO enzymes and lignin deposition in the peel tissues of ‘Jiefangzhong’ loquat fruit [24]. The defense-mediated activity of MeJA against anthracnose is also associated with higher sugar and juice contents and a reduction in internal browning, leading to the higher activities of chitinase and *β*-1,3-glucanase enzymes [99]. Moreover, the disease resistance in MeJA-treated loquat fruit might also be ascribed to the higher polyamine biosynthesis and cellular ATP levels [103]. Similar results were observed in the inhibition of anthracnose rot with the application of a 1-MCP treatment to loquat fruit [81]. Although chemical treatments have been reported to control the incidence of anthracnose rot, there may be residual levels of these chemicals that may be harmful to consumer health. Consumer concern for chemical residual effects initiated the search for biological alternatives to control the infestation of *Colletotrichum*.

Biological control is also significant in reducing the *C. gloeosporioides* infestation in loquat fruit. Yan, et al. [167] reported that there was a negative correlation between the inoculation of the *Bacillus amyloliquefaciens* MG3 strain and fungal growth, and an inhibition of spore germination. Similarly, another strain (HG_01_) of *B. amyloliquefaciens* was found to be very effective in reducing *C. acutatum* severity and delayed oxidative stress due to a lower PAL enzyme activity in loquat fruit [168]. *Bacillus cereus* AR156 induced a higher phenolic biosynthesis and H_2_O_2_ accumulation when inoculated in loquat fruit with *Colletotrichum acutatum* colonies. The possible defense-oriented pathway may be ascribed to the inhibition of membrane leakage due to higher activities of POD, PPO, PAL, *β*-1,3-glucanase and chitinase enzymes that induced higher H_2_O_2_ accumulation [169]. *Pichia membranefaciens*, a yeast, was reported to inhibit spore germination by increasing the activities of chitinase and *β*-1,3-glucanase enzymes that played a defensive role against *C. acutatum*. MeJA played a stimulating role in the enhancement of *P. memranfaciens* colonies at an exponential rate with higher biocontrol activities against the germ tube elongation of *C. acutatum* [100]. Moreover, there was also a positive role of *Pichia membranefaciens* in the reduction of *C. acutatum* spore germination and lesion diameter in loquat fruit [170].

### 7.3. Loquat Canker

Loquat is very susceptible to many bacterial diseases and cankers caused by *Pseudomonas syringae* pv. *eriobotryae* is one of the most severe diseases worldwide [171]. This disease attacks almost all parts of the plant and appears as brown lesions on the leaf and fruit surfaces that damage the mesophyll cells. There is a natural defense mechanism in loquat to alleviate canker disease, and the genetic basis of the resistance was discovered by identifying the gene locus and linkage group in the wild loquat species *Eriobotrya deflexa* [172]. Hiehata, et al. [173] outlined that the loquat resistance against *Pseudomonas syringae* can only be expressed in homozygous conditions, as *Psc-c* a recessive gene that triggers the plant defense mechanism. To check the efficacy of different hybrids obtained by two resistant cultivars, ‘Shiromogi’ and ‘Champagne’, 14 different crosses of both of these cultivars were observed, and it was found that there was more resistance among the hybrids, as compared to the original parent lines. It was further suggested that there is more than one gene involved in the disease tolerance in loquat fruit that cannot be determined by the phenotype or genotype of the *Psc-c* locus gene. Based on disease symptoms, *Pseudomonas* has been classified into three potential groups, A, B, and C, from which only group C pathogens cause brown spots on the skin of leaves and fruit [174]. However, future research on the canker control strategies in loquat on its transmission, mode of action, and control strategies in loquat fruit is warranted.

## 8. Conclusions and Future Prospects

Loquat fruit is regarded as a minor fruit containing a wide array of phytochemicals and health-promoting compounds. Short shelf life with accelerated quality degradation remains a challenge for growers, processors, and associated industry stakeholders. It is still considered an unexploited fruit crop, presenting extensive research gaps in disease control, nutrition for fruit quality, protection of on-tree delicate ripe fruit to avoid physical and mechanical injuries, postharvest handling and storage, and supply chain. Loquat is not well documented for its harvest handling and postharvest operations and for the quantitative losses in the supply chain from farmgate to consumers. During the postharvest period, low-temperature storage is coupled with certain constraints, such as susceptibility to CI, enzymatic browning, and microbial decay, which substantially downgrade the quality and marketability of stored loquat fruit.

In conclusion, loquat is highly perishable, and future research encompassing various pre- and post-harvest factors affecting postharvest quality warrants further investigations. The fruit ripening physiology of loquat fruit is worth studying, which warrants further research involving molecular approaches to determine alterations in fruit quality. The appropriate harvest maturity standards are yet to be determined for different types of storage conditions. Moreover, the integrated pre- and post-harvest approaches to managing the postharvest diseases of loquat warrant further investigation. Although cold storage is the most widely used practice for the storage of loquat, it causes CI. Considering the adverse impacts of CI on loquat fruit quality, the management strategies for CI must be considered for future research. The holistic approaches encompassing preharvest and postharvest practices impacting the storage of loquat fruit must be standardized on a genotype and regional basis to reduce postharvest losses.

## Figures and Tables

**Figure 1 foods-12-01329-f001:**
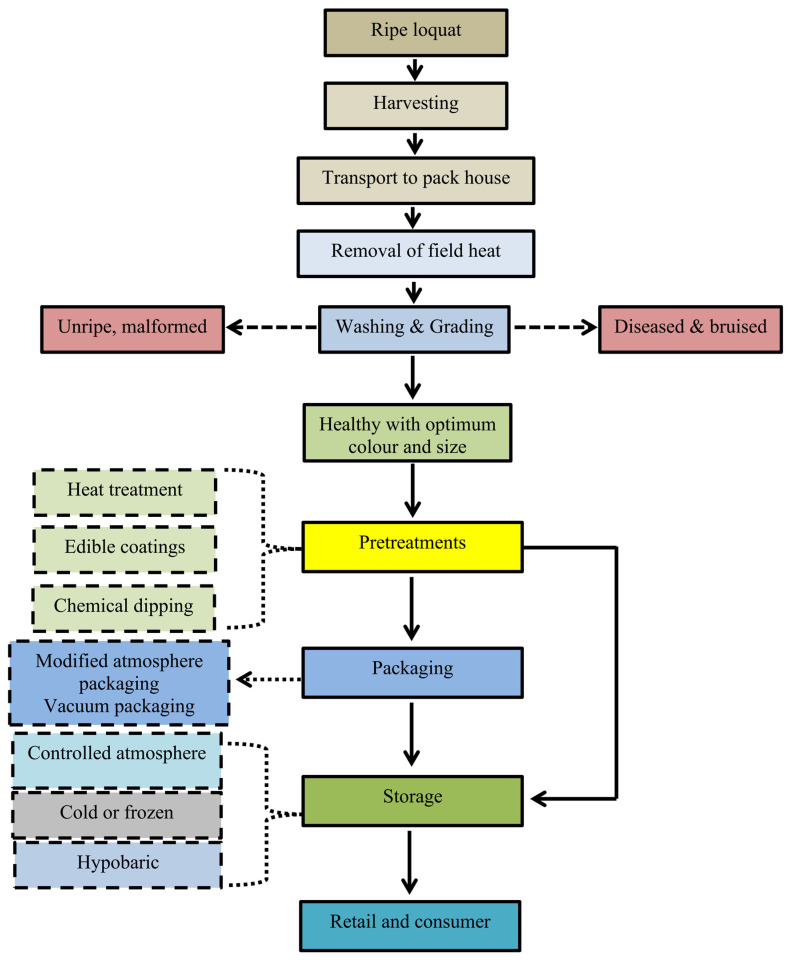
Process flow chart for the harvesting and postharvest handling of loquat fruit is described based on the literature.

**Table 1 foods-12-01329-t001:** Some important loquat cultivars grown in the world [10,11].

Cultivar	Harvest Season	Flesh Colour	Fruit Shape	Peel Colour	Fruit Flavour
Advance	Mid	White	Pear	Yellow	Acidic sweet
Ahdar	Late	White	Oval	Greenish yellow	Sweet Tart
Ahmar	Early	Yellow	Pear	Radish orange	Sweet Tart
Baiyu	Early	White	Oval	Radish blushed	Sweet to subacid
Blush	Mid	White	Pear	Yellow	Subacid
Compagne	Early	Yellowish white	Elongated pear	Golden yellow	Subacid to sweet
Dahongpao	Mid	Yellow	Oval	Orange yellow	Subacid
Fire Ball	Mid	White or straw	Ovate	Orange	Acidic
Jiefangzhong	Mid	Yellow	Oval large	Yellow	Sweet to subacid
Lyuyangqing	Early to mid	Deep yellow	Ovate	Yellowish orange	Acidic
Mammoth	Mid	Orange	Rounded oval	Orange	subacid
Matchless	Mid	Pale orange	Pear	Golden yellow	Subacid
Mogi	Early	Light yellow	Elliptical	Yellow	Sweet
Premier	Mid	White	Oblong	Salmon orange with dots	Acidic
Safeda	Early to mid	White creamy	Pear	Yellow	Acidic
Tanaka	Late	Brownish orange	Ovoid or round	Orange yellow	Sweet
Thales	Late	Orange yellow	Oblong	Yellow	Sweet
Thames Pride	Early	Pale orange	Ovate	Yellow	Subacid
Victor	Very late	White	Oblong	Deep yellow	Sweet
Zhaozong	Mid	White	Pear	Yellow	Acidic
Zhaozong No. 6	Late	Yellow	Oval	Light yellow	Sweet

**Table 2 foods-12-01329-t002:** Physical, physiological, and biochemical characteristics of loquat fruit at maturity [34,47,48,61].

Parameter	Ripe Fruit Concentration
Respiration rate at 20 °C (µL CO_2_ g^−1^ h^−1^)	36.59–48.13
Ethylene production at 20 °C (nL g^−1^ h^−1^)	0.62–1.00
Fruit firmness (N)	3.33–3.83
Lightness	61.56–63.11
a *	8.82–10.57
b *	49.64–52.05
Total soluble solids (°Brix)	7.63–12.97
Titratable acidity (%)	7.04–9.78
TSS/TA ratio	0.86–1.11
Glucose (g 100 g^−1^ FW)	1.4–1.7
Fructose (g 100 g^−1^ FW)	3.08–3.60
Sucrose (g 100 g^−1^ FW)	5.2–5.4
Malic acid (mg 100 g^−1^ FW)	101.97–150.14
Citric acid (mg 100 g^−1^ FW)	895.58–988.05
Ascorbic acid (µg g^−1^ FW)	15.7–18.3
Total phenolic content (µg GAE/g FW)	427.9–450.7
Total antioxidant content (% inhibition DPPH)	60–65
Total flavonoid content (µg rutin/g FW)	43.5–51.0
Total carotenoids (*β*-carotene/g FW)	35.7–48.3

**Table 3 foods-12-01329-t003:** Effect of heat treatment on postharvest quality management of loquat.

Cultivar	Treatment	Inference	Reference
‘Fukuhara’	Hot air treatment 37 °C for 3- and 6 h	Higher AsA, TA, and TSS content with less activity of PAL, POD, and PPO enzymes were observed. CI was suppressed.	[66]
‘Jiefangzhong’	Hot air (38 °C) for 36- and 48 h	Less fruit rot, FWL, FD, and internal browning with higher TA and TSS content. Burning symptoms were observed when stored for 48 h.	[65]
‘Jiefangzhong’	Hot air exposure (38 °C) for 5 h	Reduced MDA and H_2_O_2_ contents, higher juice content and less fruit firmness, enhanced APX, SOD, and CAT enzyme activities. Lowered LOX and superoxide radical production.	[63]
‘Jiefangzhong’	Hot air treatment (38 °C) for 5 h	Delayed the activities of PG, PPO, POD, and PAL enzymes, reduced lignin deposition and FD, maintained higher sensory quality.	[64]
‘Jiefangzhong’	Exposure to hot air at 38 °C for 5 h	Inhibition of PLD and LOX enzymes, decline in membrane leakage, higher linolenic and linoleic acid content with less stearic and oleic acid. Maintained higher unsaturated/saturated fatty acid content with less MDA content.	[62]
‘Jiefangzhong’	Hot air (38 °C) for 36 h + *Pichia guilliermondii*	*C. acutatum* was lowered with higher SOD and CAT enzymes and less ROS. Higher lignin deposition due to higher PAL enzyme.	[68]
‘Jiefangzhong’	Hot air treatment at 38 °C for 36 h	Reduced FWL, FD, internal browning, and POD, PPO, and PAL enzymes. Higher APX, CAT, and SOD enzymes with less membrane leakage. Higher TPC with less CI and ROS.	[69]
‘Jiefangzhong’	Hot air exposure (35 °C) for 3 h	Lowered MDA and H_2_O_2_ contents, maintained membrane integrity, higher NI, AI, SS, and SPS enzyme assays with less sucrose and higher fructose and glucose level. Alleviated CI symptoms.	[70]
‘Jiefangzhong’	Hot air (38 °C, 5 h) + MeJA 16 μmol L^−1^	Less protopectin, pectin, and lignin content, less PPO, POD, and PAL with higher PG, APX, CAT and SOD enzyme activities.	[71]
‘Jiefangzhong’	Hot water treatment (45 °C) for 10 min + GB 10 mmol L^−1^	Reduced MDA content and electrolyte leakage, higher CAT, SOD, and APX enzymes, higher proline, and GABA due to higher OAT, P5CS, and GAD enzymes.	[72]

FD = fruit decay, FWL = fruit weight loss, GABA = γ-aminobutyric acid, GAD = glutamate decarboxylase, GB = glycine betaine, NI = neutral invertase, OAT = ornithine δ-aminotransferase, P5CS = Δ^1^-pyrroline-5-carboxylate synthetase, RSM = response surface methodology.

**Table 4 foods-12-01329-t004:** Effect of chemical treatments on postharvest quality management of loquat fruit.

Chemical	Cultivar	Treatment	Inference	Reference
1-MCP	‘Baisha’, ‘Luoyangqing’	1 μL L^−1^ + LTC at 5 °C for 6d	Less G-POD, CAD, and PAL enzyme activities with less superoxide radicals, lignin deposition, FD at 5 °C LTC, and 1-MCP in both cultivars. ‘Luoyangqing’ was a better respondent than ‘Baisha’	[82]
	‘Claudia’	1, 2, 3, 4 and 5 μL L^−1^	CI, browning, FD, and fruit softening were delayed with better fruit sensory attributes. Best treatment was 1 μL L^−1^ 1-MCP.	[83]
	‘Claudia’ ‘Nespolone di Trabia’	0.50 and 1 μL L^−1^	Fruit firmness was maintained with better TA and lower browning in both cultivars. However, better results were presented by ‘Nespolone di Trabia’ with 1 μL L^−1^ 1-MCP treatment.	[84]
	‘Fuyang’	10, 50 and 100 nL L^−1^	CI was inhibited with a decrease in MDA, H_2_O_2_, and superoxide radicals, lower LOX, and PLC enzymes. However, CAT enzyme activity was maintained.	[79]
	‘Fuyang’	2.32 nmol L^−1^	CI was reduced. Higher linolenic and linoleic acid content, leading to higher unsaturated/saturated fatty acid ratio. Less hemicellulose and cellulose content with higher water- and CDTA-soluble pectin content.	[80]
	‘Fuyang’	50 nL L^−1^	Suppressed PAL, CAD, C4H, 4CL coenzyme A, POD, and PPO enzymes, and high PG enzyme activity. CI was also lowered. Inhibition of FD, browning, and lignin deposition were observed.	[85]
	‘Jiefangzhong’	50 nL L^−1^	Suppressed browning, FD, H_2_O_2_ content, and superoxide radicals. Improved APX, SOD, and CAT enzymes with higher chitinase and β-1,3-glucanase enzymes. Higher juice content, TSS, and TA were also exhibited. *C. acutatum* infection was inhibited.	[81]
	‘Luoyangqing’	0.5, 5 and 50 μL L^−1^	Inhibited browning and ethylene production. Lowered PPO and LOX enzymes. Maintained higher TPC and polyphenol content with reduced superoxide anion radicals.	[21]
	‘Qingzhong’	50 nL L^−1^	Lowered FD, enhanced TSS, TA, sucrose, glucose and TPC. Reduced PPO activity with higher DPPH-radical scavenging activity. Inhibited ROS production.	[78]
BTH	‘Jiefangzhong’	10, 30 and 60 mg L^−1^	Higher TSS and TA were maintained. Suppressed *C. acutatum* infestation and PAL enzyme activity. Lignin deposition was minimized, and there was aggravation of the disease tolerance mechanism.	[86]
	‘Jiefangzhong’, ‘Zhaozhong 6′	10, 30 and 60 mg L^−1^	Higher chitinase and β-1,3-glucanase, SOD, CAT, POD, and PPO enzyme activities but suppressed PAL enzyme. Reduced LOX and ROS production.	[87]
		2, 3 and 4% CaCl_2_	Maintained a lower browning index, weight loss, and TA while increasing juice content, pH, and TSS.	[88]
	‘Advance’	4% CaCl_2_ + 11 m*M* AsA + 5 m*M* CA + 5 mmol SA	Higher hue angle was exhibited. Less FWL, DI, FD, and firmness were seen. There was higher TA, TSS, and AsA content.	[89]
	‘Changhong’	1% CaCl_2_	Reduced CI, superoxide anion, H_2_O_2_, and MDA content while exhibiting higher activities of DHAR, GR, MDHAR, APX, CAT, and SOD enzymes, as well as a higher DPPH radical assay.	[90]
	‘Changhong’	1% CaCl_2,_ 10 mmol L^−1^ EGTA	Prevented CI, ion leakage, and MDA content with higher ATP, ADP, and EC. Increased Ca^2+^-ATPase and H^+^-ATPase, CCO, SDH, PAO, DAO, GAD, OAT, and P5CS enzymes. There were higher proline, polyamine, and GABA contents.	[91]
	‘Surkh’	1, 2 and 3% CaCl_2_	Reduction in EC, FWL, browning, and firmness increase. Higher TA, TSS, and AsA contents.	[23]
	‘Trouloti’	2% CaCl_2_	Sweetness was increased, and acidity declined with a lower TPC level. There was no impact on FWL and dry matter. There was a reduction in respiration rate.	[92]
	‘Qingzhong’	0.8% CaCl_2_ + 0.2, 0.4 and 0.8% PAA	DI, FD, FWL, respiration rate, and membrane leakage was decreased. Higher TSS, TA, and AsA content with better sensory attributes were observed.	[93]
Ethanol	‘Jiefangzhong’	300 μL L^−1^	Boosted activities of PAL, SOD, PPO, POD, chitinase, and β-1,3-glucanase enzymes. H_2_O_2_ was increased which initiated a defense mechanism against spores and mycelium of *C. acutatum*.	[94]
GB	‘Jiefangzhong’	10 mmol L^−1^ GB + HT 45 °C for 10 min	Higher CAT, SOD, APX, P5CS, OAT, and GAD enzymes with higher GABA and proline content. Reduction in MDA content and CI symptoms.	[72]
	‘Jiefangzhong’	1, 5, 10, 20 mmol L^−1^ GB	Higher TFC and TPC content, less FWL, and browning. Higher CAT and SOD enzymes with less MDA content. A 10 mmolL^−1^ concentration was best.	[95]
MeJA	‘Fuyang’	10 mmol L^−1^ MeJA	Reduction in respiration, ethylene production, PPO, and PAL enzymes activities. Higher total sugar and organic acid content with higher TPC and TFC was maintained. Browning and FD were lessened.	[96]
	‘Fuyang’	10 mmol L^−1^ MeJA	Reduced CI, saturated fatty acids, H_2_O_2_ content, and superoxide radicals. Fruit firmness and colour was maintained.	[97]
	‘Fuyang’	10 mmol L^−1^ MeJA	CI was lowered with less pectin, hemicellulose, pectin, alcohol content, and lignin deposition by suppressing PAL and PPO enzymes. CDTA and water soluble pectin were increased. ROS production was lowered with a remarkable decline in the POD enzyme.	[98]
	‘Jiefangzhong’	10 μmol L^−1^ MeJA	Exhibited higher level of chitinase and β-1,3-glucanase enzymes, higher TSS and TA content, less browning, FD, and spore germination of *C. acutatum*.	[24]
	‘Jiefangzhong’	10 μmol L^−1^ MeJA	Reduced lignin deposition, inhibited PPO and PAL enzyme activities. MeJA induced H_2_O_2_ content that suppressed the fungal growth of *C. acutatum* with higher CAT and APX enzymes.	[99]
	‘Jiefangzhong’	10 μmol L^−1^ MeJA + *Pichia membranefaciens* 1 × 10^8^ colony-forming units mL^−1^	Inhibited spore germination and germ tube elongation of *C. acutatum*. There was less disease incidence and higher chitinase and β-1,3-glucanase enzymes activities were maintained.	[100]
	‘Jiefangzhong’	10 μmol L^−1^ MeJA	GSH-POD, APX, and GST enzymes were enhanced with a higher AsA content. GSH content was decreased due to more MDHAR, DHAR, and GR enzyme activities, and inhibition of AO enzyme activity. Browning was also delayed.	[101]
	‘Jiefangzhong’	10 μmol L^−1^ MeJA	Higher GABA and proline content were shown with higher P5CS, OAT, and GAD enzymes and reduced PDH enzyme. CI was suppressed with less browning.	[102]
	‘Jiefangzhong’	10 μmol L^−1^ MeJA	Higher ATP, Put, Spd, and Spm content was maintained. Suppressed AMP and ADP production by increasing energy status with less anthracnose symptoms.	[103]
	‘Jiefangzhong’	MeJA 16 μmol L^−1^ + HT at 38 °C for 5 h	Less protopectin, pectin and lignin content, less PPO, POD, and PAL with higher PG, APX, CAT, and SOD enzyme activities.	[71]
	‘Luoyangqing’	10 µmol L^−1^	Induced *EjbHLH14*, *EjHB1*, and *EjPRX12* gene expression, thereby decreasing CI and lignin content.	[17]
NO	‘Luoyangqing’	0.5 mmol L^−1^ cPTIO, 0.1 mmol L^−1^ TUN, 10 m mmol L^−1^ Gln, 0.5 m mmol L^−1^, *L*-NAME and 0.5 m mmol L^−1^ PBITU	All these NO inhibitors reduced APX, CAT, POD, and SOD enzyme activities with higher MDA and H_2_O_2_ content. Membrane leakage and browning were increased. The role of NO in fruit quality has been confirmed.	[25]
Melatonin	‘Dawuxing’	50 µL melatonin	Reduced weight loss and MDA content while maintaining higher firmness, ABTS, FRAP, and DPPH radical assays	[104]
SA		1500, 2000 and 3000 ppm	Reduced browning and increased TSS, TA, and pH.	[105]
	‘Jiefangzhong’	1 gL^−1^ SA	FD, lignin deposition, and browning were decreased. Reduction in CAD, PAL, and PPO enzymes were also observed.	[106]
	‘Luoyangqing’	1 mmol L^−1^ ASA	Reduced the CI symptoms with reduced G-POD, CAD, and PAL enzyme activities and prohibited superoxide radical accumulation.	[12]
	‘Zaozhong’	40 and 70 mg L^−1^ SA	GSH was increased with higher GST, G-POD, MDHAR, DHAR, and APX enzymes, which alleviated the POD enzyme and decreased MDA and H_2_O_2_ contents with less DHA activities.	[107]
2, 4-epibrassinolide	‘Changhong’	10 µmol L^−1^ EBR	The symptoms of lignification were alleviated because of the delay in the rise in lignin concentration in loquat fruit.	[108]
Short term N_2_	‘Dahongpao’	100% N_2_	Changes in membrane permeability, MDA levels, and ROS production rates were all significantly slowed down. Furthermore, SOD and CAT activities were considerably higher, whereas LOX activity was significantly lower, in N_2_-treated fruits compared to control fruits.	[109]

ADP = adenosine diphosphate, AMP = adenosine monophosphate, AO = ascorbate oxidase, ASA = acetylsalicylic acid, ATP = adenosine triphosphate, BTH = benzothiadiazole, CA = citric acid, Ca^2+^-ATPase = Ca^2+^-adenosine triphosphate, CCO = cytochrome *c* oxidase, DAO = diamine oxidase, DHA = dehydroascorbic acid, DI = disease incidence, EC = electrical conductivity, EBR = epibrassinolide, EGTA = ethylene glycol bis (2-aminoethyl ether) tetraacetic acid, FD = fruit decay, FWL = fruit weight loss, GABA = γ-aminobutyric acid, GAD = glutamate decarboxylase, GB = glycine betaine, Gln = glutamine, GSH = glutathione, GSSG = oxidized glutathione, GST = glutathione-*S*-transferases, H^+^-ATPase = H^+^-adenosine triphosphate, H_2_O_2_ = hydrogen peroxide, HT = heat treatment, *L*-NAME = NG-nitro-l-Arg methyl ester, OAT = ornithine δ-aminotransferase, P5CS = Δ^1^-pyrroline-5-carboxylate synthetase, PAA = peracetic acid, PAO = polyamine oxidase, PBITU = S,S’-1,3-phenylene-bis(1,2-ethanediyl)-bis-isothiourea, PDH = proline dehydrogenase, PE = polyethylene, ppb = parts per billion, Put = putriscine, SA = salicylic acid, Spd = spermidine, Spm = Spermine, TUN = tungstate.

**Table 5 foods-12-01329-t005:** Effect of modified atmospheric packaging storage on postharvest quality management of loquat.

Cultivar	Treatment	Inference	Reference
‘Algerie, ‘Golden Nugget’,	PLA tray at 5 or 10 °C	Reduced browning, weight loss, and sugar content with improved organoleptic attributes.	[135]
‘Algerie’	Microperforated PE at 2 °C storage and a 4 day shelf life at 20 °C	Lowered fruit weight loss, reduced firmness increase, and maintained higher sugar and acid levels.	[29]
‘Champagne de Grasse’	(I) 312.5 ppb of 1-MCP+MAP (II) 625.5 ppb of 1-MCP+MAP	Reduced internal browning, microbial activity, better fruit colour, and higher TSS content.	[133]
‘Golden Nugget’ ‘Syeda’	12.5 µm, 14 µm, or 16 µm thick PVC films at 0 °C	Browning increased with increased PE film thickness, lowered weight loss, and firmness increase maintained higher TSS and TA content.	[136]
‘Hafif Çukurgöbek’	MAP (20 µm thick; LifePack) with 3- and 6-mM OA	Higher TPC, TFC, organic acids, and delayed browning and firmness increase was maintained, especially in 6 m*M* and MAP storage.	[132]
‘Japanese Azalea’	MAP with CA (0.5, 1.0%), AsA (1, 2%), NaHMP (0.5, 1.0%)	Higher DPPH radical scavenging activity with a better TPC, TFC, TA, TSS, and TSS/TA ratio was exhibited with less browning and weight loss.	[137]
‘Jiefangzhong’	PE packaging with 6 at 10 °C	Less respiration and fruit weight loss at 10 °C than at 6 °C with better sensory attributes.	[27]
‘Jiefangzhong’	Perforated low density PE with (4.8 ± 0.67)% CO_2_ and (11.5 ± 0.85)% O_2_.	Reduced weight loss, respiration rate, firmness increased, and maintained higher TSS, TA, and vitamin C content.	[60]
‘Karantoki’ ‘Morphitiki’	Xtend^®^ packaging + 4 °C storage + 20 °C for 2d.	‘Morphitiki’ was better in colour, TSS, TA, and higher browning was observed in ‘Karantoki’	[18]
‘Mogi’	0.15% perforated PE + 1, 5, 10, 20 and 30 °C	Suppressed respiration rate and ethylene production, reduction in malic acid and sucrose content, fruit weight loss and fruit decay were lowered.	[28]
‘Ottawianni’	MAP with (12% O_2_ + 3% CO_2_ + 85% Ar), (12% O_2_ + 3% CO_2_ + 85% N), (15% O_2_ + 5% CO_2_ + 80% Ar), (15% O_2_ + 5% CO_2_ + 80% N)	Ar-treated MAP stored fruits exhibited higher TPC and TFC content, while there was less browning in N treated MAP stored fruits.	[132]
‘Surkh’	HDPE (0.09 mm), LDPE (0.03 mm), 0.25% LDPEP at 4 °C	LDPE exhibited the least browning, HDPE maintained the lowest TSS and firmness, and LDPEP maintained the lowest TA and highest firmness.	[134]
‘Qingzhong’ ‘Dawuxing’	Nano-SiO_2_ packing (0.10% of nano-SiO_2_ with film thickness of 40 mm	Substantial reduction in the rate of internal browning, as well as slowed declines in TSS, TA, and vitamin C contents and extractable juice in both cultivars.	[138]
‘Palermitana’	PVC plastic film, transmission rates of water vapor, O_2_ and CO_2_ through CX film are 350 g m^−2^·24 h at 38 °C, 18,000 cm^3^ m^−2^·24 h,and 47,000 cm^3^ m^−2^·24 h at 23 °C.	Loquat fruits kept in cold storage and SL with film packaging significantly prevented shriveling and maintained interior quality and flavour.	[139]

Ar = argon, HDPE = high density polyethylene, LDPE = low density polyethylene, LDPEP = perforated low density polyethylene, N = nitrogen, NaHMP = sodium haxamataphosphate, OA = oxalic acid, ppb = parts per billion, PE = polyethylene, TFC = total flavonoids content.

## Data Availability

No research data is associated with this review article.

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
