# Peer review of "Postharvest Biology and Technology of Loquat (Eriobotrya japonica Lindl.)"

_foods, 2023, doi:10.3390/foods12061329_

Round 1
Reviewer 1 Report
The paper presents a review of the loquat fruit with emphasis on postharvest aspects. The paper is well organized in terms of form. In terms of content, there are some opportunities for improvement which are listed below:
- It would be good to include the scientific name in the title (Eriobotrya japonica Lindl.).
- It is important to make some mention of the different varieties, their main characteristics since in several occasions reference is made to the different varieties, which are not necessarily known by the potential readers of the document.
- In L29 revise the sentence on the effectiveness of the fruit against lipoproteins? the wording is not clear and should be improved.
- It might be interesting to add a photo of the fruit, its different stages of maturity, defects, etc.
- It is necessary to review and standardize some things, for example, in some cases it talks about Ca, others about calcium.
- The wording from L37 to L57 is convoluted.
- It is necessary to review all the numbers associated with titles and subtitles.
- In many parts the document is very qualitative, it could be improved if important numerical parameters associated with the fruit and its postharvest were added.
L168 What do you mean by decay controlling?
Revise figure 1, the third stage is not well distinguished in the text.
In terms of the chemical composition of the fruit, it would be interesting to add a chapter with the variations of this as the various possible processes are applied and post-harvest handling.
Also, there is certainly information related to the influence of the seeds in the fruit preservation process.
In this sense, it would be interesting to mention if there is any specific and unique substance of the fruit that is modified with processes and postharvest.
Standardize the use of the words cultivar and variety.
In some occasions the document contains too many abbreviations, for example in table 3, it would be good to group or look for a better way to organize the information.
The point on hypobaric management is too short and is unbalanced in relation to the rest of the treatments.
Regarding the diseases, it is necessary to mention correctly the pathogens involved in these deterioration processes.
The conclusion can be improved.
Author Response
Hi
Response to comments of Reviewer (1) has been attached.

Reviewer 2 Report
Dear author,
Thanks for your good review paper. My comments are mentioned in the attached file.
Regards

Author Response
Response to the comments of Reviewer 2 is attached.

Reviewer 3 Report
Please take a look in the attachment.

Author Response
Response to the comments of reviewer no 3 is attached.

Reviewer 4 Report
The review “Postharvest biology and technology of loquat fruit” comprehensively focuses on the recent advances in the postharvest physiology and technology of loquat fruit including harvest maturity, fruit ripening physiology, postharvest storage techniques and physiological disorders and diseases. The summarize of the postharvest physiology and technology of loquat fruit is comprehensive and systemic. However, some references are too old, and there is no schematic model that summarizes the whole text. My major comments are as follows:
1. The subheadings are not sorted correctly. Please revise them.
2. The references are too many, and some old references are able to be deleted.
3. The factors affecting ripening and postharvest of loquat (besides cold injury) need to be explained in Introduction. Moreover, the novelty of this review was not stated in the Introduction section. For example, are there any similar review recently or before? What is the significance of writing this review for actual agricultural production?
4. In this review, some terms were written inconsistently. For example, Ethylene was appeared in Page 2, Line 92. However, in Page 3, Line 105, the authors used “C2H4”. Moreover, the full name of the abbreviations should be appeared when it is firstly mentioned. For example, the full name of ACO1, ACO2 and ACS1 is missing in Page 3, Line 106. Please check the full text and revise them.
5. The contents of “FRUIT PHYSIOLOGY MODIFICATIONS DURING RIPENING”, “POSTHARVEST PHYSIOLOGICAL DISORDERS” and “DISEASES” should be summarized as Tables or Figures. In addition, there is no schematic model that summarizes the whole text.
6. The phrase between “Harvest” and “Removal of field heat” in Figure 1 was missing.
7. The conclusions of the determine harvest maturity, physiological changes during fruit ripening should be added. Additionally, the future work is missing in CONCLUSIONS AND FUTURE PROSPECTS section.
Author Response
Response to comments attached.

Round 2
Reviewer 1 Report
Dear Editor
The authors have provided satisfactory responses to this reviewer's comments and it is recommended that the paper be accepted in its present form.
Reviewer 2 Report
Dear author,
Many thanks for the revised manuscript.
Regards
Reviewer 4 Report
Thank you for your careful revision.